# Significant increase in eco-efficiency of China's grain production from 2000 to 2022: Trend changes, typological evolution, and driving factors

Huali Jin[1,2]*, Chao Han[1]

**1** School of Economics, Shandong University of Finance and Economics, Jinan, China, **2** Center for High Quality Development, Shandong University of Finance and Economics, Jinan, China

* 15188279905@163.com

## Abstract

Enhancing the eco-efficiency of grain production is a critical avenue for ensuring food security and ecological sustainability. This study employs a global super-efficiency SBM model incorporating undesirable outputs, combined with the life cycle assessment method, to comprehensively measure the eco-efficiency of grain production in 31 Chinese provinces and municipalities from 2000 to 2022. Furthermore, we conduct a comprehensive analysis of the distributional dynamics and key driving factors of the eco-efficiency of grain production. The findings indicate that: (1) The overall level of eco-efficiency in China's grain production is relatively low, exhibiting significant regional disparities. The spatial pattern follows the gradient of "major grain-producing regions> production-sales balance regions> the major grain-consuming regions," with most provinces yet to reach the efficiency frontier. (2) The eco-efficiency of grain production in China generally exhibits an upward trend, although there are indications of spatial polarization, evident "club convergence" characteristics, and a notable "positive spillover" effect. (3) The eco-efficiency of grain production in China is influenced by a complex interplay of factors, including economic, social, technological, demographic, and natural elements. The gross total agricultural output, water resources endowment, and structure of agricultural production emerge as the critical driving factors, manifesting the Matthew effect of "the rich getting richer and the poor getting poorer." The findings of this study provide a foundation for the refinement of sustainable grain production policies and the promotion of green agricultural transformation.

## 1. Introduction

The challenges of food security and ecological sustainability pose formidable barriers to global sustainable development. As the planet's most populous emerging

**Data availability statement:** The author has uploaded the data, and the DOI required to access this data is: https://doi.org/10.6084/m9.figshare.30069364.v1.

**Funding:** This work was funded by the National Social Science Foundation of China (NSSFC) Project "Research on the Coordinated Promotion of Agricultural Emission Reduction, Carbon Sequestration, and Food Security under the Background of 'Double Carbon' Goals" (No. 22AJY008).

**Competing interests:** The authors declare that they have no known competing financial interests or personal relationships that could have appeared to influence the work reported in this paper.

economy, China sustains 22% of the world's inhabitants with merely 9% of the Earth's cultivable terrain [1]. Regarding the trio of principal cereal crops—rice, wheat, and maize—China's aggregate yearly yield constitutes roughly 30%, 18%, and 21% of global production, respectively [2], thus making a crucial contribution to worldwide food security. Nevertheless, swift industrialization and urban expansion have exposed China's grain production to unparalleled resource and ecological strains [3,4]. On one hand, the per capital tillable land is steadily shrinking, and the overuse of agrochemicals has resulted in increasingly acute problems like soil deterioration and diffuse pollution [5]. China's fertilizer consumption per unit area has surpassed twice the ceiling established by the United Nations Food and Agriculture Organization (FAO), while pesticide usage exceeds 2.5 times the global norm [6]. Conversely, China's farming-related greenhouse gas output represents about 14% of the nation's total emissions, with grain cultivation being the dominant contributor [7]. In the face of global climate shifts and sustainable agricultural advancement, the conventional productivity-centric approach to grain production demands urgent renovation and enhancement [8].

In response to these challenges, this study aims to evaluate the current status and evolutionary patterns of China's agricultural eco-efficiency metrics, while conducting a thorough analysis of the key factors driving its transformation. Previous academic research has demonstrated that optimizing agronomic practices and innovating nutrient management are crucial pathways for improving ecological efficiency. For instance, Asdullah et al. (2024) found that exogenous melatonin application reduces post-harvest losses [9], while Ahmed et al. (2022) revealed significant variations in the distribution of soil available nitrogen and associated environmental impacts under different fertilization strategies [10]. These studies provide valuable insights for enhancing ecological efficiency in agricultural production. This research adopts a macro-level perspective to examine the dynamic evolutionary patterns of ecological efficiency and their driving mechanisms, which not only promotes green and low-carbon development in Chinese agriculture and ensures national food security and ecological safety, but also contributes to China's approach to addressing global challenges in sustainable agricultural development.

The concept of ecological efficiency emerged from sustainable development theory and was formally introduced by the World Business Council for Sustainable Development (WBCSD) in the 1990s, with its core principle defined as "creating more economic value with less resource consumption and environmental impact." Serving as a critical bridge between economic growth and environmental protection, ecological efficiency has become a key instrument for achieving the United Nations Sustainable Development Goals (UN SDGs). As global sustainability challenges become increasingly severe, both theoretical research and practical applications of this concept have demonstrated a rapidly growing trend. Scholars have delved into the internal characteristics of eco-efficiency [11], the goal requirements for eco-efficiency [12], the realization paths for eco-efficiency [13], and other detailed theoretical discussions. Moreover, they have developed index systems to quantitatively assess eco-efficiency based on its theoretical implications [14,15]. As research

progresses, scholars have begun to examine eco-efficiency issues in various fields, including arable land [16], farming [17], livestock [18], the marine industry [19], agriculture [20], industry [21], and services [22]. Among these, eco-friendly advancement of cereal cultivation, a crucial area related to national livelihood, has garnered significant attention.

Early studies on grain production primarily concentrated on supply quantity and price stabilization [23]. With the emergence of sustainable development principles, scholars have progressively recognized that development models solely pursuing yield increases while neglecting resource and environmental costs are no longer viable. The research focus in food production has shifted from traditional yield-oriented approaches to resource and environmental efficiency orientation, with the ecological efficiency of food production increasingly gaining scholarly attention [8].

Within the sustainable development framework, research on ecological efficiency in food production has evolved from a single-dimensional approach to multi-dimensional integration. In recent years, scholars have conducted a series of studies examining carbon emission efficiency in food production [24], environmental efficiency of food production [25], and green total factor productivity in food systems [26], while exploring their patio-temporal heterogeneity. This evolution reflects the theoretical deepening process of sustainable food production research from considering single environmental factors to comprehensive assessment of multiple ecological and environmental dimensions. These theoretical explorations and empirical studies have significantly enriched the research content on sustainable food production development and laid the foundation for constructing a more comprehensive ecological efficiency evaluation system for food production.

Among the various indicators used to assess the sustainable development of grain production, eco-efficiency has garnered significant attention for its ability to comprehensively evaluate both economic output and environmental impact. Now, the primary methods for assessing the eco-efficiency of grain production include LCA (Life Cycle Assessment), SFA (Stochastic Frontier Analysis) and DEA (Data Envelopment Analysis). LCA is an effective method for measuring the environmental impact of the entire grain production process, with a particular emphasis on carbon emissions [27]. Data Envelopment Analysis (DEA) models are widely employed due to their ability to handle multiple inputs and outputs without the need for a specific production function form [28]. However, researchers employing DEA models to assess grain production diverge in their selection of undesirable output indicators. While the majority of scholars consider surface source pollution in grain production to be an undesirable output [29], only a limited number of studies have included carbon emissions as an undesired output [26]. In light of the pressing need to reduce greenhouse gas emissions in agriculture, it is crucial to consider both surface source pollution and carbon emissions in the grain production process when evaluating eco-efficiency.

China's resource endowments exhibit considerable variation across provinces, and the factors influencing the eco-efficiency of grain production are complex and multifaceted. Over time, China's Five-Year Plans have strategically guided the grain production sector from a focus on increasing yields to incorporating sustainable development considerations [30]. This shift in policy orientation has theoretically resulted in a gradual evolution of the drivers of eco-efficiency in grain production [31]. With regard to spatial comparison, China categorizes provinces into three distinct categories: major grain-producing regions, the major grain-consuming regions, and production-sales balance regions. These categories are based on interprovincial resource endowments, climatic conditions, and grain production and marketing dynamics. As the regional division of labor in grain production deepens, the drivers of eco-efficiency in grain production also exhibit regional variations in geospatial terms [32]. However, the majority of existing studies are limited to the three primary regional divisions in the East, Central, and West of China. Few of these studies analyze the dynamic evolution characteristics of grain production eco-efficiency and its drivers based on vertical and horizontal comparative perspectives, while taking into account the unique characteristics of China's grain production functional areas. Moreover, existing research has primarily employed various statistical methods, including ordinary least squares [33], spatial Durbin model [34,35], and Tobit regression [36], to identify the drivers of eco-efficiency in grain production based on regional averages. This approach fails to consider the discrepancies in drivers at different efficiency levels, which ultimately leads to conclusions with limited applicability and policy relevance.

The existing literature has deepened the understanding of eco-efficiency in grain production, but research in this domain remains at a preliminary stage. A comprehensive assessment system that integrates the global super-efficiency SBM model and the LCA method has not yet been established, nor have the dynamic evolutionary characteristics and driving factors been thoroughly investigated. In view of these research gaps, the marginal contributions of the present study are manifested in the following three aspects: First, the study innovatively integrates the global super-efficiency SBM model and the LCA method to assess the eco-efficiency of grain production. By constructing a comprehensive efficiency measurement system that accounts for non-point source pollution and carbon emissions, it provides a comprehensive evaluation of the current status and improvement potential of the eco-efficiency of grain production in China. Second, the study analyzes the distributional trends and spatial interactions of China's grain production eco-efficiency in dynamic evolution characteristics and long-term transfer trends. This provides a scientific foundation for enhancing grain production eco-efficiency. Finally, the study employs the quantile regression method to identify the key drivers of eco-efficiency in grain production in China across different efficiency levels in the dimensions of the whole, time period, and region. This provides decision-makers with the requisite information to advance green and sustainable development in grain production while ensuring the security of local food supplies.

## 2. Materials and methods

### 2.1. Life-cycle assessment methodology

This paper establishes an accounting approach for carbon emissions from grain production throughout the entire life cycle of "input factors—planting management—growth and development—straw combustion," based on the research concept of the life cycle assessment method [37]. The methodology involves the following steps:

First, carbon emissions attributable to production inputs. The cultivation of grains involves the application of various agrochemicals and materials, such as synthetic fertilizers, crop protection agents, and polyethylene films, all of which contribute directly to $CO_2$ emissions during the production process. The emission factors are based on the standards outlined by the IPCC, https://www.ipcc.ch/.

Second, carbon emissions caused by planting management are taken into account. Activities such as plowing and irrigation in the process of grain cultivation also result in direct $CO_2$ emissions, with corresponding carbon emission coefficients sourced from the IPCC.

Third, Greenhouse gas emissions from crop physiological processes. Biological activities during cereal plant growth significantly contribute to atmospheric methane ($CH_4$) and nitro oxide ($N_2O$) release. These non-$CO_2$ emissions originate from soil-plant interactions in agricultural ecosystems. It is noteworthy that methane emissions from rice paddies represent the most significant source of carbon emissions in grain production [7]. China's rice paddy area represents 30% of the global total, thereby underscoring the country's pivotal role in worldwide $CH_4$ emissions [2,38]. Conversely, dryland soils in arid ecosystems exhibit a methane uptake function, resulting in minimal methane emissions from these environments [39]. Consequently, the present study focused exclusively on $CH_4$ emissions during the growth and development of rice, employing the $CH_4$ emission coefficients for distinct rice varieties during their fertility periods, as provided by Tian and Zhang (2013) [40]. These coefficients incorporate the influence of fertilizer application on $CH_4$ emissions from rice fields, thereby eliminating the necessity to independently consider $CH_4$ emissions stemming from fertilizer use.

Fourth, the burning of straw contributes to carbon emissions. Following the harvesting of grain, the open burning of straw releases various pollutants, including $CO_2$, $N_2O$, and $CH_4$. The emission factors for crop residue incineration were derived from the findings of Li et al. (2016) [41]. The specific measurement procedure is outlined below:

$$CO_{2input} = A_c \times \delta_c + B_c \times \delta_c + C_c \times \delta_c + D_c \times \delta_c + E_c \times \delta_c + F_c \times \delta_c$$
$$CH_{4rice} = \sum G_c \times \delta_c$$
$$\begin{cases} CO_{2straw} = \sum H_c \times I_c \times J_c \times L_c \times K_c \times \delta_c \\ N_2O_{straw} = \sum H_c \times I_c \times J_c \times L_c \times K_c \times \delta_c \\ CH_{4straw} = \sum H_c \times I_c \times J_c \times L_c \times K_c \times \delta_c \end{cases}$$
$$C = CO_{2input} + \alpha_1 \times CH_{4rice} + CO_{2straw} + \alpha_2 \times N_2O_{straw} + \alpha_1 \times CH_{4straw}$$

(1)

In the formula, $\delta c$ represents the carbon source coefficient. $CO_{2input}$ denotes the $CO_2$ emissions caused by input factors and farming practices. $A_c$, $B_c$, $C_c$, and $D_c$ represent the quantities of fertilizer, pesticide, agricultural plastic film, and diesel fuel utilized, respectively. $E_c$ and Fc indicate the area sown with grain crops and the effectively irrigated area of grain crops, respectively. $CH_{4rice}$ represents the methane generated during the growth and development of rice. $G_c$ represents the cultivated areas for early, middle, and late rice, respectively. $CO_{2straw}$, $N_2O_{straw}$, and $CH_{4straw}$ denote the $CO_2$, $N_2O$, and $CH_4$ produced from straw burning, respectively. $H_c$, $I_c$, $J_c$, $K_c$, and $L_c$ represent various grain yields, grain-to-straw ratios, straw dry matter ratios, combustion efficiencies, and combustion ratios, respectively. Finally, C denotes the aggregate greenhouse gas emissions associated with cereal cultivation. To facilitate the calculation, this study converts all greenhouse gases to standard $CO_2$. $\alpha$ and $\beta$ are the coefficients utilized to convert all pollutants to $CO_2$ emissions. In accordance with the IPCC standard, the $CO_2$ conversion coefficients for $CH_4$ and $N_2O$ are 25 and 298, respectively.

## 2.2. Global super-efficient SBM model

This study employs a global super-efficiency Slacks-Based Measure (SBM) model, incorporating undesirable outputs, to assess grain production eco-efficiency. The methodology, based on Tone (2001) [42], involves two steps: Calculating efficiency scores using the global standard efficiency SBM model with undesirable outputs for all decision-making units (DMUs). Applying the global super-efficiency SBM model with undesirable outputs for effective DMUs. Our research focuses on this global super-efficiency SBM model, which accounts for non-expected outputs in grain production.

Assuming that the efficiency values of n DMUs are to be evaluated, for any $DMU_j$ (j = 1,2,...,n) in period t (t = 1,2,...,T), all of them are co-produced using m inputs x=$(x_1, x_2,...,x_m)^T \in Rm^+$, and co-produced with q outputs y=$(y_1, y_2,...,yq)^T \in R_q^+$ and h kinds of undesired outputs b=$(b_1, b_2,...,b_h)^T \in R_h^+$. Drawing on the work of Huang et al. (2014) [43], the following mathematical formulation quantifies the eco-efficiency of cereal cultivation.

$$\rho^* = \min_{\lambda, s^-, s^+} \frac{1 + \frac{1}{m} \sum_{i=1}^{m} \frac{s_{io}^{-,\tau}}{x_{io}^{\tau}}}{1 - \frac{1}{q+h} \left( \sum_{r=1}^{q} \frac{s_{ro}^{+,\tau}}{y_{ro}^{\tau}} + \sum_{k=1}^{h} \frac{s_{ko}^{-,\tau}}{b_{ko}^{\tau}} \right)}$$

$$s.t.\ x_{io}^{\tau} \geq \sum_{t=1}^{T} \sum_{j=1(j \neq o \text{if} t=\tau)}^{n} \lambda_j^t x_{ij}^t - s_{io}^{-,\tau}; i = 1,2,\ldots,m;$$

$$y_{ro}^{\tau} \leq \sum_{t=1}^{T} \sum_{j=1(j \neq o \text{if} t=\tau)}^{n} \lambda_j^t y_{rj}^t + s_{ro}^{+,\tau}; r = 1,2,\ldots,q;$$

$$b_{ko}^{\tau} \geq \sum_{t=1}^{T} \sum_{j=1(j \neq o \text{if} t=\tau)}^{n} \lambda_j^t b_{kj}^t - s_{ko}^{-,\tau}; k = 1,2,\ldots,h;$$

$$1 - \frac{1}{q+h} \left( \sum_{r=1}^{q} \frac{s_{ro}^{+,\tau}}{y_{ro}^{\tau}} + \sum_{k=1}^{h} \frac{s_{ko}^{-,\tau}}{b_{ko}^{\tau}} \right) \geq \varepsilon; \lambda_j^t \geq 0;$$

$$s_{io}^{-,\tau}, s_{ro}^{+,\tau}, s_{ko}^{-,\tau} \geq 0; t = 1,2,\ldots,T; j = 1,2,\ldots,n (j \neq o \text{if} t = \tau)$$

(2)

In the formula above, $\rho^*$ represents the efficiency score of the decision-making unit (DMU), with its economic interpretation being the proportion of input resources that can be reduced while maintaining the current output level. Specifically, the efficiency score $\rho^*$ derived from this model has a range of $\rho^* \geq 1$. If $\rho^* = 1$, it indicates that the DMU is positioned on the optimal production frontier and is technically efficient. If $\rho^* < 1$, it suggests that technical inefficiency exists within

the DMU; the lower the efficiency score, the higher the degree of technical inefficiency and the greater the potential for improvement. If $\rho^* > 1$ (the super-efficiency case), it demonstrates that the DMU is located beyond the production frontier, with its technical efficiency exceeding the average level among the sample.

The efficiency scores derived from this model carry significant practical implications. Taking the efficiency score of 0.682 as an example, this indicates that China's grain production systems could, on average, reduce input resources (such as fertilizers, pesticides, water resources, etc.) by 31.8% while maintaining current grain output levels. The economic interpretations across different efficiency intervals are as follows: For scores < 0.5: This signifies severe resource waste and negative environmental externalities in the region's grain production system, with potential for at least a 50% reduction in resource inputs and pollution emissions; For $0.5 \leq$ scores < 0.8: This represents moderate efficiency levels, with substantial room for improvement in resource utilization and environmental protection; For $0.8 \leq$ scores < 1: This indicates relatively high efficiency levels, though minor improvements are still possible; For scores = 1: This demonstrates complete technical efficiency, with resource utilization reaching optimal levels; For scores > 1: This reflects a super-efficiency case, where the region's production technology and management practices outperform those of other regions.

This research achieves a triple methodological breakthrough by integrating Life Cycle Assessment (LCA) with a Global Super-efficiency SBM model: Firstly, the introduction of LCA addresses the deficiency of traditional DEA models in quantifying undesirable outputs such as non-point source pollution and carbon emissions, establishing a "sowing-to-harvest" full-chain environmental footprint assessment system that enables comprehensive quantification of ecological impacts throughout the entire grain production process, rather than merely focusing on localized environmental effects of specific stages. Secondly, efficiency measurements with global reference overcome the temporal segment limitations of traditional window analysis, ensuring dynamic comparability of cross-period data spanning 2000–2022, thus avoiding inconsistencies in efficiency evaluation standards across different time periods. Finally, environmentally-adjusted super-efficiency scores enable precise identification of technically inefficient decision-making units, providing a targeted basis for differentiated policy formulation. This integration significantly transcends the research paradigms in existing literature that employ single models or partial improvements, achieving a triple advancement: from single environmental factors to full life cycle in assessment dimensions; from static cross-sections to dynamic evolution in temporal scope; and from general recommendations to differentiated guidance in policy precision.

### 2.3. Methods for analyzing the evolution of spatio-temporal dynamics

Kernel density estimation (KDE) is a non-parametric technique visualizing the dynamic evolution of grain production eco-efficiency distribution. The traditional KDE method primarily analyzes spatial distribution of non-equilibrium phenomena. The spatial KDE incorporates geographic factors, revealing long-term regional transfer trends. This study applies spatial KDE to agricultural eco-efficiency analysis. Using grain production eco-efficiency values, we categorize each region and its neighbors into four distinct groups, providing insights into spatial eco-efficiency patterns. Regions exhibiting a ratio below 25% of the highest-level region in the country are classified as exhibiting low efficiency. Regions exhibiting a ratio between 25% and 50% are classified as exhibiting medium-low efficiency, while those exhibiting a ratio between 50% and 75% are classified as exhibiting medium-high efficiency. Regions with a ratio exceeding 75% are considered to exhibit high efficiency. Equation (3) defines the conventional kernel density estimation, while its spatial counterpart is formulated in equation (4).

$$f(x) = \frac{1}{nh} \sum_{i=1}^{n} K\left(\frac{x_i - x}{h}\right)$$
$$K(x) = \frac{1}{\sqrt{2\pi}} \exp\left(-\frac{x^2}{2}\right) \tag{3}$$

$$f(x,y) = \frac{1}{nh_xh_y} \sum_{i=1}^{n} Kx\left(\frac{X_i-x}{h_x}\right)Ky\left(\frac{Y_i-y}{h_y}\right)$$
$$g(y|x) = \frac{f(x,y)}{f(x)}$$

(4)

In equation (3), f(x) denotes the probability density function, where xi represents the provincial cereal production eco-efficiency, and xi signifies its national mean. The variable n indicates the number of regions studied, h is the bandwidth parameter, and K symbolizes the Gaussian kernel function. Equation (4) introduces f(x,y) as the bivariate probability density, with g(y|x) representing the spatially conditioned kernel density.

## 2.4. Drivers assessment methodology

(1) Model Selection: To elucidate the primary determinants of cereal cultivation eco-efficiency at different performance levels, we implemented a quantile regression model, building upon the foundational research of Koenker and Bassett (1978) [44].

$$\ln Y_{it,q} = \alpha_{i,q} + \sum_{k=1}^{K} \beta_{k,q} X_{itk,q} + u_{it,q}$$

(5)

In the equation, the letter i represents the region, the temporal dimension is denoted by t, while q represents the quantile. $LnY_{it,q}$ symbolizes the natural logarithm of cereal cultivation eco-efficiency for China's ith administrative region in year t at the qth quantile. While $X_{itk,q}$ represents the explanatory variables. The coefficient of influence of the explanatory variables is denoted by β, α stands for the fixed effect, and u represents the residual term. The spatial kernel density analysis categorizes the grain production eco-efficiency of all 31 provinces and cities into four grades based on their average size. Consequently, this study will perform a regression analysis at three quantiles: The analysis will be conducted at three quantiles: 25%, 50%, and 75%. This approach permits the examination of the impact of each influencing factor on grain production eco-efficiency at different critical points.

(2) Variable Selection: The dependent variable, $Y_i$, represents the eco-efficiency of grain production, which is evaluated using the global super-efficiency SBM model.

The explanatory variables, represented by Xi, comprise 18 indicators. As previously discussed, the factors influencing grain production eco-efficiency are multifaceted and can be examined from diverse perspectives. These factors can be categorized into five dimensions: economic status, living standards, production conditions, agricultural workforce, and natural resources. Table 1 provides a comprehensive overview of the indicator variables and the rationale for their inclusion in the study.

## 2.5. Definition of variables and data processing

### 2.5.1. Input indicators.
To ensure the consistency of data acquisition, weighting coefficients were developed to differentiate grain production input indicators from broader agricultural activities. The weighting coefficients, α and β, are defined as follows:

α = (value of agricultural output/ total value of agriculture, forestry, animal husbandry, and fishery output) × (sown area of food crops/ total sown area of crops)

β = sown area of food crops/ total sown area of crops

The study incorporates seven input indicators: (1) Land input: quantified by food crop cultivation area; (2) Labor input: calculated as primary sector employment multiplied by coefficient β; (3) Fertilizer input: Estimated by application volume

**Table 1. Variable selection and descriptive statistics.**

| First-level indicators | Second-level indicators | Symbol | Unit | Mean | Standard Deviation | Selection criteria |
|---|---|---|---|---|---|---|
| eco-efficiency | Eco-efficiency of grain production | EGP | – | 0.478 | 0.179 | – |
| Economic level | Economic growth | EG | CNY per capita | 30758.510 | 28202.920 | [45] |
| | Gross total agricultural output value | GAOV | 100 million CNY | 1334.334 | 1269.049 | [3] |
| | Fiscal support for agriculture | FSA | 100 million CNY | 0.096 | 0.041 | [46] |
| Living standard | Urbanization rate | UR | % | 52.223 | 16.184 | [47] |
| | Urban-rural income gap | URIR | – | 2.999 | 0.611 | [48] |
| | Income level of rural residents | IR | CNY | 9100.729 | 6886.895 | [48,49] |
| | Consumption level of rural residents | CR | CNY | 7279.666 | 5309.839 | [48,49] |
| Technological progress | Level of agricultural science and technology investment | ASTI | 100 million CNY | 0.941 | 5.281 | [50] |
| | Level of agricultural mechanization | AM | 10,000 kilowatts | 2829.164 | 2717.864 | [50] |
| | Level of agricultural informatization | AI | 10,000 kilometers | 2.634 | 1.688 | [50] |
| Agricultural laborers | Number of agricultural laborers | ALN | 10,000 people | 904.560 | 707.403 | [49] |
| | Quality of agricultural laborers | ALQ | year | 9.391 | 1.696 | [49] |
| | Degree of rural ageing | DRA | % | 13.520 | 4.242 | [51] |
| | Rural population density | RPD | – | 0.706 | 1.089 | [49] |
| Natural endowment | Area of soil and water conservation | SWC | 1,000 hectares | 3581.991 | 3004.090 | [52] |
| | Natural disasters | ND | – | 0.215 | 0.161 | [50,52] |
| | Water resources endowment | WRE | m³/person | 6856.527 | 26964.470 | [52] |
| | Structure of agricultural production | APS | – | 0.661 | 0.134 | [3,53] |

adjusted with coefficient β; (4) Pesticide input: measured through pesticide application, modified by coefficient β; (5) Agricultural film input: measured by the amount of plastic film used for agricultural purposes multiplied by the coefficient β; (6) Irrigation input: determined by the effective irrigated area multiplied by the coefficient β; (7) Machinery input: assessed via aggregate farm equipment capacity, adjusted by coefficient β.

**2.5.2. Output indicators.** Desired outputs: measured by grain production and carbon sequestration resulting from food crop cultivation. (1) Quantification of grain production: Grain production is quantified by the total amount of food produced by agricultural producers within a calendar year. (2) Estimation of carbon sequestration by food crops: The dynamic process of atmospheric $CO_2$ fixation through photosynthesis by food crops (including seven categories: rice, wheat, maize, millet, sorghum, potatoes, and legumes) is considered. The estimation formula is as follows:

$$Cs = \sum_{i=1}^{n} Csi = \sum_{i=1}^{n} \left\{ [Ci \times Yi \times (1 - Wi)] / Hi \right\}$$

(6)

In this equation, Cs denotes the aggregate carbon sequestration by cereal crops. The subscript i signifies specific crop varieties. Parameters $C_i$, $Y_i$, $W_i$, and $H_i$ represent the carbon content ratio, economic yield, moisture factor, and economic coefficient, respectively. These crop-specific parameters were derived from Tian and Zhang (2013) [40].

Negative externalities encompass: (1) Greenhouse gas emissions: Evaluated using LCA methodology, as previously outlined. (2) Diffuse agricultural pollutants: Assessed following the framework developed by Lai et al. (2004) [54]. Diffuse agrochemical runoff from cereal cultivation is evaluated using a standardized field-level analysis protocol. The unit pollution coefficients of fertilizers in farmland are determined based on their chemical composition. Nutrient release coefficients vary by fertilizer type, influencing the estimation of Total Nitrogen (TN) and Total Phosphorus (TP) runoff. For TN, N-fertilizers, P-fertilizers, and NPK-compound (1:1:1) have coefficients of 1.0, 0.0, and 0.33 respectively.

Correspondingly, TP coefficients are 0.0, 0.44, and 0.15. These values inform the runoff estimation equations: TN = (N-fert × 1.0 + NPK × 0.33) × leaching factor, and TP = (P-fert × 0.44 + NPK × 0.15) × leaching factor. In these equations, N-fert, P-fert, and NPK denote the application rates of respective fertilizers, while the leaching factor accounts for site-specific nutrient loss conditions.

**2.5.3. Discussion of data limitations.** While the data used in this study were derived from reliable sources, certain limitations exist, primarily manifested in two aspects: First, the issue of regional heterogeneity in emission factors. Although we explicitly utilized authoritative emission factors provided by IPCC, Tian and Zhang (2013), and Li et al. (2016) [40,41], the application of these coefficients across national scales may overlook natural environmental variations between regions. For instance, differences in soil physicochemical properties between northern and southern regions (such as organic matter content, pH value, etc.) may lead to variations in emissions under identical inputs; temperature and humidity differences across climate zones (temperate, subtropical, tropical) affect the formation mechanisms of $CH_4$ and $N_2O$, thereby influencing the accuracy of emission factors. During the study period, agricultural technological advancements may have altered certain emission factors. For example, the promotion of novel slow-release fertilizers may have reduced carbon emission intensity per unit of chemical fertilizer; updates in rice varieties and improvements in irrigation technologies may have affected $CH_4$ emission factors. The fixed emission factors employed in this study may underestimate the contribution of technological progress to emission reduction. Second, the accuracy issue with long-term provincial data series. This research utilized 23-year provincial panel data spanning 2000–2022. Throughout this period, China's statistical system underwent multiple adjustments, with changes in statistical standards and calculation methodologies for certain indicators, potentially affecting the longitudinal comparability of the data.

## 3. Spatial and temporal features of grain production eco-efficiency in China

### 3.1. Overall characteristics

China's eco-efficiency in grain production exhibited a positive trajectory, escalating from 0.383 at the turn of the millennium to 0.682 by 2022. However, it has not yet reached the optimal efficiency frontier, suggesting a potential for further improvement of 31.8% (Fig 1). An examination of the trend across sub-periods reveals distinct stage characteristics in China's grain production eco-efficiency (Fig 2). The highest average eco-efficiency was observed during the 14th Five-Year Plan period, reaching 0.663, followed by the 13th and 12th Five-Year Plan periods, with average values of 0.550 and 0.490, respectively. The 11th and 15th Five-Year Plan periods had lower average efficiency values of 0.420 and 0.398, respectively.

Multiple factors contribute to these observed patterns. China's cereal cultivation system is notable for its resource-intensive practices, which, while boosting yields, often result in substantial environmental externalities and suboptimal resource utilization. China's agrochemical efficacy lags behind developed nations, leading to elevated greenhouse gas emissions and non-point source pollution [55]. This phenomenon jeopardizes the long-term viability of cereal production. In response, China has implemented a series of policy initiatives: the 12th Five-Year Plan focused on mitigating agricultural diffuse pollution; the 13th emphasized eco-friendly farming practices; and the 14th advocates for a sustainable, low-carbon circular economy. These strategic measures have encouraged farmers to adopt environmentally conscious methods, gradually enhancing the ecological performance of cereal cultivation.

### 3.2. Regional differences

Longitudinal analysis revealed fluctuating yet upward trajectories in the environmental performance index of cereal cultivation across principal production zones, major consumption areas, and regions with balanced output and demand. These distinct categories exhibited compound annual growth rates of 3.39%, 2.43%, and 1.85%, respectively, over the

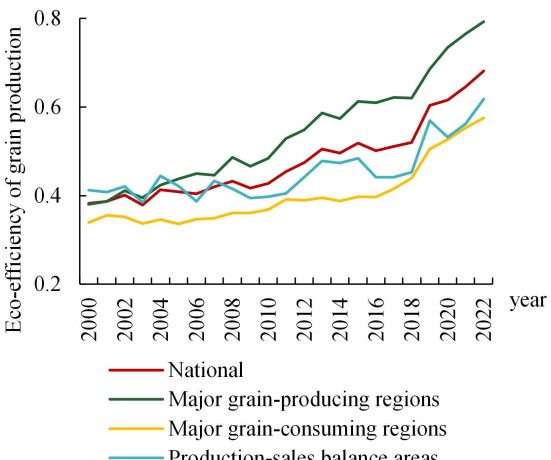

**Fig 1. Trend of changes in eco-efficiency of grain production.**

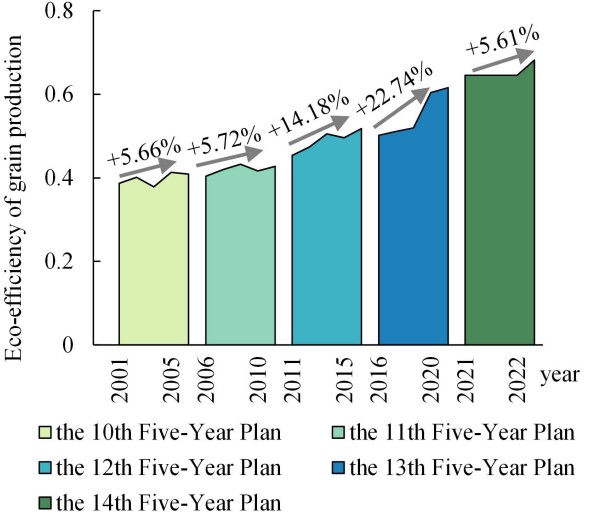

**Fig 2. Trends of changes in eco-efficiency of grain production by sub-periods.**

study period (Fig 1). Spatially, the spatial pattern of cereal production eco-efficiency has changed from the previous pattern of "production-sales balance regions> major grain-producing regions> the major grain-consuming regions" to the current pattern of "major grain-producing regions> production-sales balance regions> the major grain-consuming regions".

The main reasons for this shift can be attributed to the following factors: (1) China has implemented policies targeting major grain-producing regions, which have not only achieved long-term stable growth in grain production, but also effectively reduced carbon emissions and surface pollution associated with grain production. In particular, the highly concentrated and large-scale fertilizer reduction and efficiency improvement initiatives in key production areas have promoted the scale effect of green and low-carbon grain production. (2) In 2001, market-oriented reforms in cereal procurement and distribution led to a reduction in cultivated acreage within primary production zones. Concurrently, these regions prioritized

industrial and service sector growth, neglecting agricultural advancement and impeding eco-efficiency gains. Conversely, sustained policy support bolstered primary production areas, while structural imbalances persisted in major consumption regions. These divergent trajectories resulted in heterogeneous spatiotemporal patterns of environmental performance across the three regional categories in cereal cultivation.

## 3.3. Inter-provincial differences

Interprovincial analysis reveals a consistent upward trajectory in the environmental performance index of China's cereal cultivation, accompanied by significant regional heterogeneity. (1) Except for Hainan and Tibet, all other provinces have experienced positive growth. The five provinces with the highest growth rates are Liaoning, Inner Mongolia, Tianjin, Heilongjiang and Jilin, with average annual growth rates ranging from 4% to 6%. Except for Tianjin, the eco-efficiency values of these provinces exceed 1 in 2022, reaching the efficiency frontier. These principal cereal-producing provinces have allocated substantial resources towards advancing "dual-focus agriculture" (emphasizing quality and efficiency) and eco-friendly cultivation practices. Such initiatives have paved the way for enduring progress in sustainable cereal production. (2) Conversely, the five provinces with the slowest growth rates are Tibet, Hainan, Guizhou, Guangxi and Guangdong. The efficiency scores of these provinces remain below 0.5 in 2022, suggesting that there is more than 50% potential for improvement. This substantial regional variation poses a considerable obstacle to enhancing the nationwide environmental performance index in cereal cultivation.

## 3.4. Comparative case studies of evolutionary patterns in production systems across typical regions

### 3.4.1. Case study of major production area: analysis of transformation characteristics in Heilongjiang Province.
As a strategic core area for national food security, Heilongjiang Province demonstrates a transformation pathway characterized by significant policy-technology-scale synergistic driving forces: First, leveraging key projects such as the Sanjiang Plain Agricultural Comprehensive Development, the province has established a trinity foundational support system integrating farmland protection, soil fertility enhancement, and water conservancy safeguards. Second, through its agricultural reclamation system, the province has constructed a comprehensive "R&D-extension-application" technology diffusion mechanism, substantially increasing the contribution rate of agricultural scientific and technological advancements. Third, by developing new-type agribusiness entities with scaled operations, the province has achieved enhanced efficiency in resource-intensive utilization, maintained relatively low fertilizer and pesticide application intensity per unit area, and demonstrated a synergistic development pathway that balances grain production increases with ecological protection.

### 3.4.2. Case study of major consumption area: analysis of development constraints in Guangdong Province.
With Guangdong Province's grain self-sufficiency rate declining to 25.4%, its ecological efficiency reflects the transformation dilemma facing highly urbanized regions: On one hand, the compression effect from industrialization and urbanization has intensified the industrialization-urbanization paradox, exacerbating both diseconomies of scale and negative externalities in grain production, while small-scale farmers face constraints such as high adoption barriers for new technologies and insufficient incentives for green production practices. Examining the root causes reveals that: first, reforms in farmland property rights have lagged behind, and market allocation mechanisms for urban-rural construction land remain imperfect, resulting in the continuous loss of premium cropland; second, agricultural support policy instruments remain overly singular, failing to effectively address the structural contradictions characterized by small-scale, high-cost, and low-yield production.

### 3.4.3. Case study of production-consumption balanced area: exploring transformation models in Guangxi Province.
The innovative value of Guangxi Province lies in exploring integration pathways between moderate-scale operations with circular agriculture: First, the province has constructed a "sugarcane-livestock farming-compost recycling" material circulation system, significantly improving nutrient cycling efficiency within agricultural production systems.

Second, the province has developed county-level characteristic agricultural industrial clusters (such as selenium-rich rice industrial belts), enhancing value-added per unit of resource through industrial chain extension. Third, the province has implemented ecological agricultural models in karst landform areas, promoting three-dimensional agroforestry-pastoral systems combining "fruit trees-forage grass-livestock" in rocky desertification control zones, achieving synergistic improvement in both resource carrying capacity and production efficiency in ecologically vulnerable regions.

## 4. Dynamic evolution of eco-efficiency of grain production in China

### 4.1. Evolution of the distributional dynamics of eco-efficiency in China's grain production

Analysis of distributional dynamics (Fig 3) reveals a pronounced rightward shift in China's overall environmental performance index for cereal cultivation, with varying intensities across regions. Principal production zones exhibit the most substantial progression, followed by major consumption areas, while balanced production-sales regions show the least advancement. This pattern of improvement corroborates earlier findings on regional eco-efficiency trends in cereal cultivation. The observed rightward movement, albeit of differing magnitudes, is consistent across all regional categories, indicating a broad enhancement in environmental performance within the agricultural sector. The primary driving force behind this phenomenon is likely the fact that each region, leveraging its resource advantages and policy support, has implemented and promoted a series of green varieties and technologies in accordance with the development philosophy of "resource utilization, minimization, and colonization." By focusing on the core principles of "weight reduction, pesticide control, field cleanliness, restoration, and recycling," these regions have successfully demonstrated and popularized these practices, thereby facilitating the transformation of grain production from a solely supply- and income-oriented endeavor to one that prioritizes pollution reduction, emission mitigation, and the enhancement of green practices and carbon sequestration.

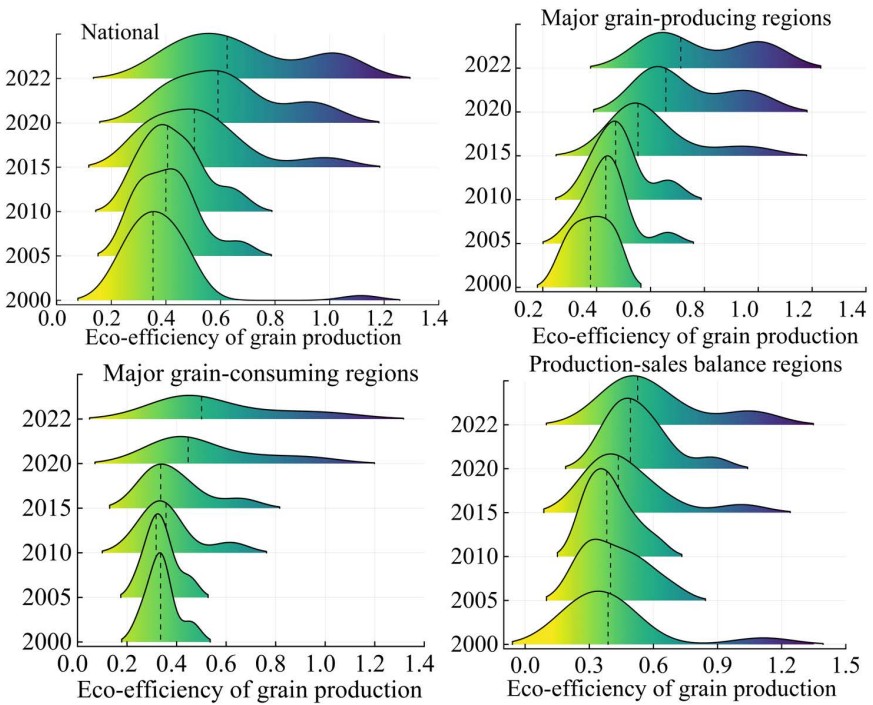

**Fig 3. Conventional kernel density plot of eco-efficiency of grain production in China.**

Examining polarization characteristics (Fig 3), the distribution of China's cereal cultivation environmental performance index exhibits a bimodal pattern, comprising a primary and a secondary peak. The primary peak demonstrates a consistent decrease in amplitude coupled with an increase in breadth, indicating a bifurcation into high and low performance clusters. However, this polarization trend attenuates over time. Notably, the three major functional zones display marked heterogeneity in their environmental performance distributions, reflecting diverse regional dynamics in agricultural sustainability. The major grain-producing regions remain consistent with the overall national distribution characteristics, while the internal differentiation within the major grain-consuming regions gradually increases. In the production-sales balance regions, the initial intensification of the internal polarization phenomenon is followed by a subsequent weakening over time.

### 4.2. Long-term shifting trends in the eco-efficiency of grain production in China

**4.2.1. Unconditional kernel density estimation.** Findings from the unconstrained kernel density analysis (Fig 4) indicate that China's environmental performance index for cereal cultivation primarily clusters along the positive 45° diagonal. This indicates a strong persistence in eco-efficiency from year t to year t+1 across provinces, with relatively low levels of mobility. This finding suggests the potential existence of "club convergence." In regions with lower eco-efficiency, there is a discernible upward shift in grain production eco-efficiency from year t to year t+1, implying some improvement in the developmental conditions of these lower-performing provinces. Conversely, in regions with higher eco-efficiency, the distribution of transfer probability density exhibits a more pronounced deviation from the diagonal, particularly in provinces where grain production eco-efficiency surpasses 1.0. In provinces with high eco-efficiency, the transfer probability density is significantly lower than the 45° diagonal, indicating the presence of developmental bottlenecks. Provinces including Heilongjiang, Jilin, Liaoning, and Shanghai demonstrate a tendency towards future decline in their cereal cultivation environmental performance index.

Sub-regional analysis (Fig 5) reveals a distinctive "club convergence" pattern in the environmental performance index for cereal cultivation across all three functional zones. This convergence may be attributed to the similarity in initial resource endowments, geographic locations, and development patterns among provinces within each region, which has led to a gradual alignment over time [56].

In lower-efficiency areas, the transitional probability density for both primary cereal production and consumption zones predominantly flanks the 45° diagonal. This distribution pattern suggests minimal fluctuations in the environmental performance index for cereal cultivation within these regions. Conversely, in higher-performing areas (e.g., approaching 0.9), the transitional probability density distribution notably diverges from the 45° diagonal, indicating a tendency towards decline

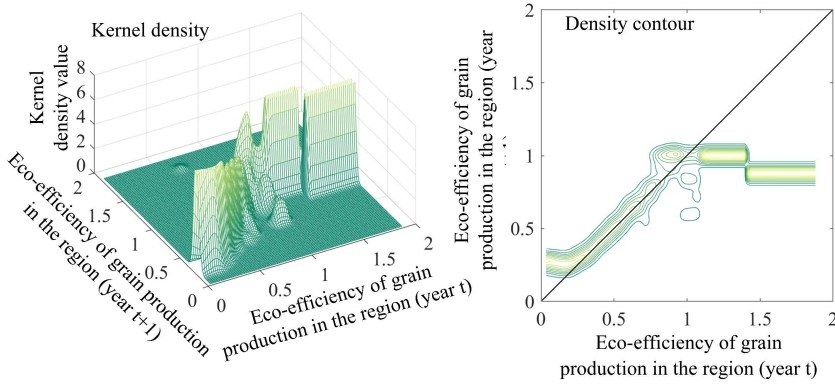

**Fig 4. Unconditional kernel density and density contours of eco-efficiency of grain production in China.**

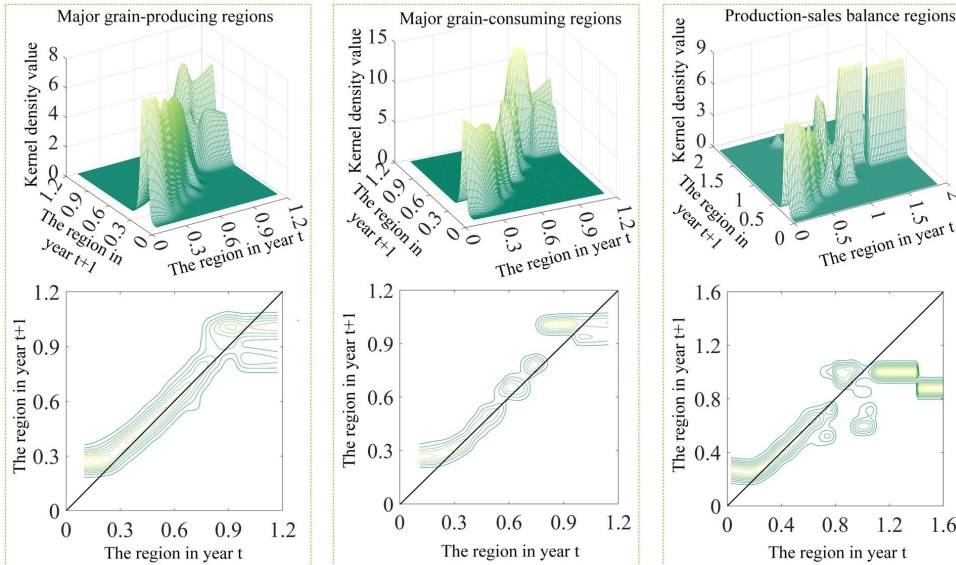

**Fig 5. Unconditional kernel density and density contours of eco-efficiency of grain production by region.**

in cereal cultivation environmental performance among these top-tier provinces. The distribution of transfer probability density in the production-sales balance regions exhibits a distinct discontinuity phenomenon, with 0.8 serving as a critical threshold point delineating divergent evolutionary trajectories. When the eco-efficiency of grain production falls below 0.8, the transfer probability density is distributed evenly along both sides of the 45° diagonal. Conversely, when it exceeds 0.8, the transfer probability density is observed to skew below the 45° diagonal, indicating a propensity for high-efficiency regions to experience a decline in efficiency levels within a one-year timeframe.

**4.2.2. Dynamic Kernel density estimation for spatial conditions.** From a holistic perspective (Fig 6), when considering spatial factors from a dynamic viewpoint, China's grain production eco-efficiency exhibits a significant positive spillover effect. When neighboring regions demonstrate low efficiency in year t, the conditional probability body remains roughly parallel to the X-axis, indicating the absence of a notable spatial spillover effect under these circumstances. When adjacent areas display moderate performance levels in year t, the probability mass gravitates towards the positive 45°

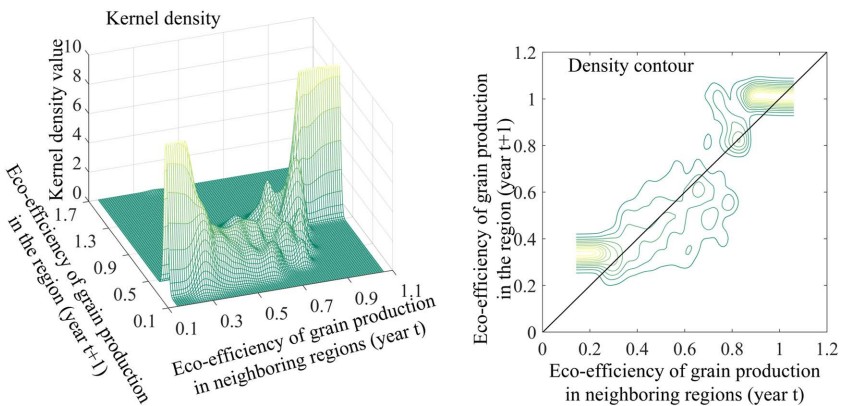

**Fig 6. Spatial conditional dynamic kernel density and density contours of eco-efficiency of grain production in China.**

diagonal, suggesting a beneficial spatial externality. Furthermore, in cases where neighboring regions exhibit superior performance in year t, the probability concentration predominantly lies above the positive 45° diagonal. This distribution pattern indicates that high-performing adjacent areas exert a substantial positive externality on the focal region, characterized by a notable magnitude of spillover effects.

Sub-regional analysis (Fig 7) reveals that the kernel density patterns of cereal cultivation environmental performance in primary production and consumption zones largely mirror those observed at the aggregate level. Adjacent regions exhibiting moderate to superior performance levels demonstrate notable positive externalities, suggesting that enhanced environmental efficiency in neighboring areas correlates with improved outcomes in the focal zone. Conversely, the probability body in production-sales balance regions tends to align closely with the X-axis, suggesting a weak spatial spillover effect in these areas.

Further analysis, using a threshold of 0.8 for grain production eco-efficiency in neighboring regions within the production-sales balance regions, revealed that when the grain production eco-efficiency in neighboring regions in year t falls below 0.8, the probability distribution shifts towards the positive 45° diagonal, implying the presence of a positive spatial spillover effect. When adjacent areas achieve environmental performance indices exceeding 0.8, the probability distribution demonstrates a more pronounced horizontal alignment, diverging from patterns observed in unconditioned kernel density analyses. This parallel alignment is concentrated within the interval of 0.4 to 0.6 on the Y-axis and is shifted downward, indicating that the correlation between neighboring regions with grain production eco-efficiency above 0.8 and the focal region weakens when a one-year lag is introduced.

Spatial spillover effects primarily originate through three major transmission pathways: First, technology diffusion effects, whereby conservation tillage techniques and precision fertilization techniques developed in major grain-producing regions spread to surrounding areas through agricultural technology extension systems. For example, black soil conservation techniques from Heilongjiang's agricultural reclamation system have been successfully promoted to neighboring provinces such as Jilin and Liaoning, generating significant technological spillovers. Second, policy emulation effects, primarily manifested as successful policy practices from high-efficiency regions being adapted and adopted by surrounding areas. For instance, Jiangsu Province's high-standard farmland construction practices have been rapidly emulated by adjacent provinces like Anhui and Zhejiang, forming regional policy learning networks. Third, factor mobility effects, whereby the

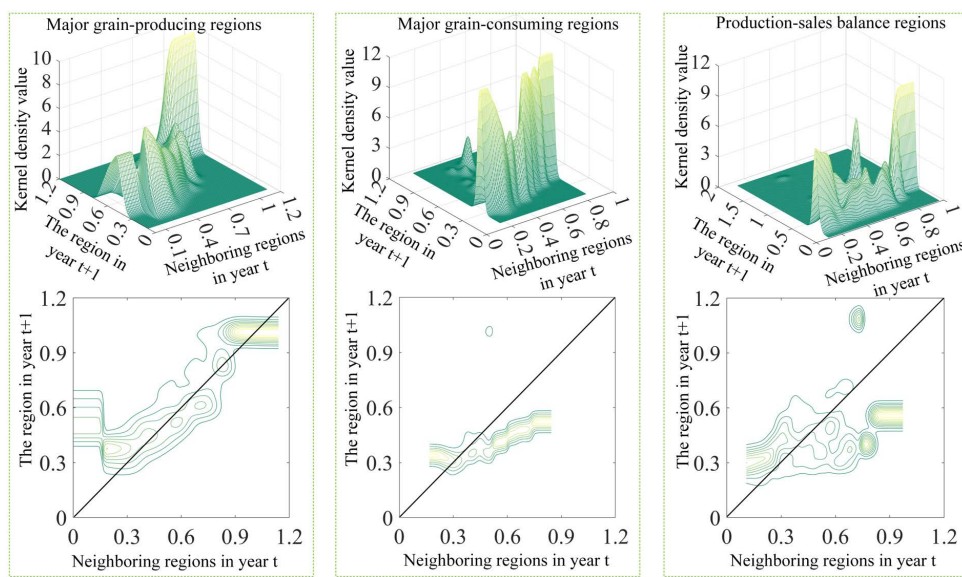

**Fig 7. Spatial conditional dynamic kernel density and density contours of eco-efficiency of grain production by region.**

movement of new-type professional farmers and entrepreneurs across regions facilitates the transmission of management expertise and production models, promoting the diffusion of environmentally friendly agricultural practices. Empirical studies indicate that these spillovers exhibit a clear "distance-decay" pattern, with high-efficiency regions typically forming "growth poles" for efficiency enhancement, whose influence diminishes as distance increases. This finding offers spatial-scale insights for constructing regional collaborative mechanisms, and accounts for why certain adjacent provinces exhibit patterns of "club convergence" in ecological efficiency.

## 5. Spatial and temporal patterns of eco-efficiency of grain production in China and the drivers of its evolution

This segment investigates the primary factors influencing cereal cultivation environmental performance across various efficiency strata, considering both developmental phases and regional attributes of China's agricultural sector. From a longitudinal perspective, China's "Five-Year" development plans serve as temporal nodes, allowing for the investigation of the changes in eco-efficiency drivers across various development stages. This approach is designed to elucidate the characteristics and patterns of these drivers over time. From a horizontal perspective, the three principal functional zones of grain production are employed as spatial units to analyze the disparities in drivers across regions. This investigation facilitates the development of tailored strategies for improving regional agricultural environmental performance, considering the unique circumstances and needs of individual localities.

### 5.1. Corporate level

A comprehensive analysis reveals eight key determinants influencing cereal cultivation environmental performance across various efficiency levels. These include economic expansion, aggregate agricultural production value, urban-rural income disparity, agricultural workforce size, rural population aging, soil and water preservation area, water resource availability, and agricultural production structure. Among these factors, four demonstrate notable inverse correlations with agricultural environmental efficiency: economic expansion, urban-rural income disparity, agricultural workforce size, and soil and water preservation area. Conversely, four factors exhibit positive associations: aggregate agricultural production value, rural population aging, water resource availability, and agricultural production structure.

The impact of related factors on grain production eco-efficiency varies significantly across different levels of eco-efficiency (Table 2). Firstly, the regression coefficient of the structure of agricultural production is significantly positive, displaying a "U-shaped" pattern as the quantile point increases. Initially, there is a decrease, followed by an increase. This suggests that optimizing the agricultural production structure is more beneficial for enhancing grain production eco-efficiency in low--efficiency and high-efficiency regions. Nevertheless, the efficacy of these measures may be diminished in medium-efficiency regions. Secondly, the regression coefficient of the degree of rural aging is significantly positive, exhibiting an "inverted U-shape" trend as the quantile point rises, initially increasing and then decreasing. This indicates that the aging of the rural population exerts the most pronounced positive impact on grain production eco-efficiency in medium-efficiency regions, while the impact is relatively weaker in regions with lower or higher efficiency levels.

The analysis reveals that the regression coefficient for aggregate agricultural production value exhibits a significant positive trend, intensifying with rising quantile points. This suggests that enhancing overall agricultural output positively influences cereal cultivation's environmental performance across various efficiency strata. Conversely, the coefficient representing urban-rural income disparity demonstrates a significant negative trend, becoming more pronounced as quantile points ascend. This implies that the detrimental impact of widening income gaps between urban and rural areas on agricultural environmental efficiency is particularly acute in regions with lower performance levels.

**Table 2. Regression results of factors influencing grain production eco-efficiency at the national aggregate level.**

| First-level indicators | Second-level indicators | q=25% | q=50% | q=75% |
|---|---|---|---|---|
| Economic level | EGP | −0.301*** | −0.321*** | −0.263*** |
| | EG | 0.082** | 0.267*** | 0.280*** |
| | GAOV | 0.053 | 0.104** | 0.002 |
| Living standard | FSA | 0.391*** | 0.002 | −0.176* |
| | UR | −0.373*** | −0.319** | −0.195* |
| | URIR | 0.191 | 0.230 | 0.328** |
| | IR | −0.134 | −0.267* | −0.161 |
| Technological progress | CR | 0.022** | 0.012 | −0.029** |
| | ASTI | 0.066** | −0.042 | −0.025 |
| | AM | 0.140*** | 0.110*** | 0.023 |
| Agricultural labor force | AI | −0.150*** | −0.303*** | −0.356*** |
| | ALN | −0.137 | 0.24 | 0.169 |
| | ALQ | 0.246*** | 0.272*** | 0.186*** |
| | DRA | −0.025 | −0.056* | −0.0005 |
| Natural endowment | RPD | −0.043*** | −0.027*** | −0.030*** |
| | SWC | −0.036*** | −0.018 | 0.003 |
| | ND | 0.062*** | 0.039*** | 0.100*** |
| | WRE | 0.407*** | 0.335*** | 0.584*** |
| Constant | C | 0.0443 | 2.467*** | 0.783 |

Note: *, **, *** denote significance at the 10%, 5%, and 1% levels, respectively.

## 5.2. Time dimension

Table 3 highlights notable variations in key factors influencing cereal cultivation environmental performance across different time frames. During the 10th Five-Year Plan, the main determinants were predominantly linked to natural resource endowments. The 12th Five-Year Plan saw the emergence of additional influential elements, including aggregate agricultural output value, governmental agricultural support, rural consumption levels, agricultural labor quality, rural aging trends, and rural population density. As China entered the 14th Five-Year Plan period, the primary catalysts for agricultural environmental efficiency shifted towards progressive domains. This evolution indicates a transformation in China's agricultural sector, transitioning from a resource-intensive model dependent on natural conditions to an eco-friendly, carbon-efficient, and sustainable approach. This new paradigm emphasizes enhanced resource utilization efficiency and reduced environmental impact.

## 5.3. Regional level

Table 4 illustrates that a wide range of factors, including economic level, living standard, technological progress, agricultural loborers, and natural endowment considerations, significantly influence the eco-efficiency of grain production across all three functional areas. Among these factors, the level of agricultural science and technology investment stands out as a particularly important driver, showing a significant effect across the three main functional areas. It is, however, important to note that the mechanisms and extent of influence of each driver differ across the three functional areas at various efficiency levels.

In primary cereal cultivation zones, regions exhibiting lower environmental performance (q=25%) are influenced by 10 significant factors, while those with moderate efficiency levels (q=50%) are affected by 12 key determinants. These

**Table 3. Regression results of factors affecting eco-efficiency of grain production by sub-period.**

| Period | The 10th Five-Year Plan period | | | The 12th Five-Year Plan period | | | The 14th Five-Year Plan period | | |
|---|---|---|---|---|---|---|---|---|---|
| Variables | q = 25% | q = 50% | q = 75% | q = 25% | q = 50% | q = 75% | q = 25% | q = 50% | q = 75% |
| EGP | −1.009*** | −0.814*** | −0.529*** | −0.136 | −0.242 | −0.396*** | 0.298* | −0.391 | −0.220 |
| EG | 0.091 | 0.134 | 0.098 | −0.005 | 0.017 | 0.447*** | −0.397*** | −0.233 | −0.364* |
| GAOV | −0.088 | −0.096 | −0.015 | 0.557*** | 0.481* | 0.288 | 0.054 | −0.007 | −0.158 |
| FSA | 0.784*** | 0.254 | 0.056 | 0.905*** | 0.369 | 0.645** | −1.232** | −2.075** | −1.761 |
| UR | 0.029 | −0.061 | −0.335* | −1.273*** | −1.190** | −0.935** | −1.336*** | −0.585 | −1.230* |
| URIR | 0.966** | 0.849** | 0.522* | −0.578* | −0.554 | −0.385 | −0.729** | −0.224 | 0.092 |
| IR | −0.434 | −0.285 | −0.286 | −0.514* | −0.463 | −0.112 | −0.219 | −0.081 | −0.607 |
| CR | 0.027 | 0.088*** | 0.069*** | −0.020 | 0.011 | −0.024 | −0.065** | −0.133*** | −0.163*** |
| ASTI | 0.125 | 0.053 | 0.034 | 0.138 | 0.213 | 0.069 | 0.448*** | 0.573*** | 0.491*** |
| AM | 0.135** | 0.068 | 0.117*** | 0.399*** | 0.317*** | −0.003 | 0.214*** | 0.175** | 0.158* |
| AI | −0.224* | −0.244** | −0.242*** | −0.284** | −0.361** | −0.575*** | −0.024 | −0.204 | −0.094 |
| ALN | −0.325 | −0.515* | −0.283 | −0.265 | 0.0570 | 0.806** | 1.005*** | 0.447 | 0.014 |
| ALQ | −0.071 | 0.045 | 0.139 | 0.829*** | 0.716*** | 0.329** | 0.491*** | 0.479* | 0.519* |
| DRA | 0.014 | −0.008 | −0.016 | −0.672* | −0.938** | −0.076 | −0.987*** | −2.522*** | −1.988*** |
| RPD | −0.056*** | −0.029** | −0.036*** | −0.030** | −0.008 | −0.021 | −0.032*** | −0.014 | −0.008 |
| SWC | −0.058 | −0.110*** | −0.097*** | −0.056** | −0.035 | −0.036 | 0.019 | −0.007 | 0.029 |
| ND | 0.122*** | 0.151*** | 0.141*** | 0.030 | 0.041 | 0.108*** | −0.039 | 0.028 | 0.038 |
| WRE | −0.161 | 0.205 | 0.416*** | −0.162 | −0.363 | 0.032 | −0.365** | −0.430 | −0.488 |
| Constant | 1.045 | 1.772 | 2.867** | 7.537*** | 8.947** | 3.658 | 6.735** | 8.488 | 10.18 |

Note: *, **, *** denote significance at the 10%, 5%, and 1% levels, respectively. Due to space limitations, the results for the 11th and 13th Five-Year Plan periods are not listed.

drivers span across multiple dimensions, including economic level, living standard, technological progress, agricultural labor, and natural endowment. This indicates that grain production in these regions requires comprehensive support and inputs. In contrast, the high-efficiency region (q = 75%) has 8 significant drivers, but notably lacks drivers related to natural endowment. Key influencing elements predominantly focus on technological advancements and production inputs. These encompass expenditure on agronomic research, farm equipment adoption rates, digital agriculture implementation, and agricultural workforce size.

In the major grain-consuming regions, the areas with low eco-efficiency of grain production (q = 25%) have 9 significant drivers. In the medium-efficiency regions (q = 50%), the number of significant drivers increases to 10, with the addition of the level of agricultural information and the number of agricultural laborers, while the influence of rural population density decreases. In the high-efficiency regions (q = 75%), the number of significant drivers declines to 8, with reduced significance of three indicators: gross total agricultural output value, income level of rural residents, and the level of agricultural informatization. However, the impact of the level of agricultural mechanization has increases.

In the production-sales balance regions, the areas with low eco-efficiency of grain production (q = 25%) have 12 significant drivers. In the medium-efficiency regions (q = 50%), the 12 significant drivers remain largely unchanged, except for a decreased impact of water resources endowment and an increased importance of the consumption level of rural residents. Zones exhibiting superior environmental efficiency (q = 75%) are influenced by fewer key factors, namely six. These encompass aggregate agricultural production value, urban development index, agronomic R&D investment, farm mechanization rate, agricultural labor force, and land conservation area.

**Table 4. Regression results of factors affecting eco-efficiency of grain production at the regional level.**

| Region | Major grain-producing regions, | | | Major grain-consuming regions | | | Production-sales balance regions | | |
|---|---|---|---|---|---|---|---|---|---|
| Variables | q=25% | q=50% | q=75% | q=25% | q=50% | q=75% | q=25% | q=50% | q=75% |
| EGP | −0.418*** | −0.405*** | −0.536*** | −0.015 | −0.026 | 0.075 | −0.032 | −0.155 | −0.209 |
| EG | 0.232*** | 0.255*** | 0.201** | 0.106** | 0.106** | 0.059 | 0.154*** | 0.119*** | 0.277*** |
| GAOV | −0.003 | 0.003 | −0.001 | −0.029 | −0.001 | 0.030 | −0.246*** | −0.224*** | −0.162 |
| FSA | 0.085 | −0.144 | −0.126 | −0.492** | −0.708*** | −0.839*** | 0.265* | 0.643*** | 0.717** |
| UR | 0.259 | −0.040 | 0.045 | 0.188 | 0.195 | 0.047 | 0.755*** | 0.589*** | 0.326 |
| URIR | 1.156*** | 1.406*** | 1.246*** | 0.301* | 0.319* | 0.071 | 0.495*** | 0.729*** | 0.221 |
| IR | −0.725*** | −0.958*** | −0.731*** | −0.173 | −0.178 | 0.0000287 | −0.180 | −0.392** | −0.075 |
| CR | −0.055*** | −0.063*** | −0.051** | 0.052*** | 0.039** | 0.039** | −0.051*** | −0.035** | −0.061*** |
| ASTI | −0.174*** | −0.121*** | −0.133*** | −0.007 | 0.011 | 0.096* | −0.334*** | −0.384*** | −0.307*** |
| AM | 0.126** | 0.174*** | 0.211*** | 0.024 | 0.074** | 0.044 | −0.030 | 0.047 | 0.006 |
| AI | −0.318*** | −0.379*** | −0.390*** | −0.056 | −0.110** | −0.090* | −0.046 | −0.010 | −0.176** |
| ALN | 0.323 | 0.209 | 0.318 | −0.215 | −0.178 | 0.107 | −0.0303 | −0.165 | 0.138 |
| ALQ | 0.085 | 0.028 | 0.079 | 0.221*** | 0.225*** | 0.185*** | 0.136** | 0.183** | 0.072 |
| DRA | −0.026 | −0.047** | −0.013 | 0.041** | 0.015 | −0.003 | 0.073*** | 0.064** | 0.056 |
| RPD | −0.053** | −0.042*** | −0.011 | −0.029*** | −0.034*** | −0.044*** | −0.038** | −0.051** | −0.085** |
| SWC | −0.032 | −0.016 | −0.013 | −0.012 | −0.006 | 0.010 | −0.019 | 0.0002 | 0.006 |
| ND | 0.114*** | 0.145*** | 0.028 | −0.048** | −0.083*** | −0.077*** | −0.039** | −0.013 | 0.054 |
| WRE | −0.00003 | −0.00006*** | −0.000006 | 0.232*** | 0.271*** | 0.399*** | −0.581*** | −0.429*** | 0.021 |
| Constant | −0.878 | −0.0200 | 1.364 | 0.045 | 1.398 | 1.019 | −4.302*** | −3.873*** | −2.369 |

Note: *, **, *** denote significance at the 10%, 5%, and 1% levels, respectively.

## 6. Conclusions and recommendations

### 6.1. Conclusion

This research utilizes a combined approach of global super-efficiency SBM modeling and life cycle assessment (LCA) to analyze cereal cultivation environmental performance across 31 Chinese provinces from 2000 to 2022. By synthesizing conventional and spatial kernel density estimation techniques, the study investigates the dynamic patterns and long-term shifts in agricultural eco-efficiency distribution. The analysis identifies critical factors influencing environmental efficiency at various performance levels, considering holistic, temporal, and spatial dimensions. The findings indicate that:(1) The overall eco-efficiency level of grain production in China is relatively low, with a 31.8% potential for improvement and significant regional disparities following a spatial pattern of "major grain-producing regions> production-sales balance regions> the major grain-consuming regions" (2) While grain production eco-efficiency generally shows improvement, spatial polarization is evident. This is demonstrated by the gradual weakening of polarization in the major grain-producing regions, expansion in the major grain-consuming regions, and initial strengthening followed by weakening in the production-sales balance regions. (3) The eco-efficiency of grain production is characterized by "club convergence" and a "positive spillover" effect. (4) The primary drivers influencing grain production eco-efficiency evolve from natural endowment to scion-economic factors and then to technological progress. Across all levels of efficiency, gross total agricultural output value, water resources endowment, and structure of agricultural production consistently exhibit a significant positive impact. This phenomenon underscores the cumulative advantage principle, wherein resource-rich entities progressively enhance their position, while resource-poor ones experience further decline. (5) This study, through the innovative integration of Life Cycle Assessment (LCA) and Global Super-efficiency SBM model, not only extends the research perspective on ecological efficiency evaluation of grain production, but also provides more scientifically robust methodological

support for cross-temporal, multi-regional dynamic efficiency comparisons. The findings of this study hold significant implications for China's agricultural strategy; in the context of dual-carbon goals and rural revitalization, enhancing the ecological efficiency of grain production emerges as pivotal to agricultural green transformation. The "club convergence" and "positive spillovers" identified in this research provide theoretical foundations for constructing regional collaborative mechanisms.

### 6.2. Recommendations

The findings of the research study lead to the following recommendations:

First, Categorized guidance and region-specific implementation. (a) Major Production Areas: Leveraging green scale effects and consolidating ecological efficiency advantages. Establish agricultural technology integration demonstration bases in Heilongjiang, Jilin, and other regions; promote precision fertilization techniques; implement large-scale black soil conservation tillage; and establish crop rotation and fallow systems, allocating appropriate quantities of cropland for fallowing. (b) Major Consumption Areas: Developing high-quality, high-efficiency agriculture and accelerating ecological transformation. Construct several urban modern agricultural demonstration zones in Guangdong, Zhejiang, and other regions to improve output efficiency per unit area. (c) Production-Consumption Balanced Areas: Developing specialty ecological agriculture and constructing circular agricultural systems. Develop multiple specialty agricultural demonstration parks in Guangxi, Yunnan, and other regions to increase the growth rate of output value from specialty agricultural products.

Second, Regional coordination for holistic improvement. Establish a regional coordination mechanism led by the Ministry of Agriculture and Rural Affairs; convene regular joint conferences on collaborative development among the three functional area types; establish dedicated funds for regional ecological efficiency collaboration; and incorporate grain production ecological efficiency into government performance evaluations. Form a national expert committee on enhancing grain production ecological efficiency; implement the "Science and Technology Commissioner Plus" initiative, deploying numerous technical personnel to serve on the front lines; establish an agricultural technology training system to conduct large-scale training programs for new-type professional farmers.

Third, policy coordination for systematic advancement. Establish a green and low-carbon agriculture subsidy system that incorporates reduction and efficiency enhancement into subsidy criteria; reform agricultural insurance policies by developing green crop insurance products; and establish an agricultural carbon sink trading mechanism. Continuously expand the area through high-efficiency water-saving irrigation; construct multiple regional grain logistics centers; and develop numerous rural clean energy demonstration villages. Establish a national green agriculture development fund; develop green agricultural credit products; and encourage the issuance of specialized green agricultural bonds. Implementation of these recommendations would likely ensure national food security while simultaneously reducing agricultural carbon emission intensity, improving utilization rates of fertilizers and pesticides, decreasing agricultural non-point source pollutant emissions, and providing robust support for green and low-carbon agricultural development.

Despite the meaningful findings generated by this study, certain limitations remain. First, the assumption of nationally uniform emission coefficients may underestimate the effects of regional heterogeneity. Second, long time-series data may be subject to inconsistencies in statistical methodologies and measurement errors. Finally, provincial-level analysis may mask efficiency disparities among different areas within provinces. Future research could enhance study quality by developing regionally differentiated emission coefficient systems and employing finer-grained data at municipal and county levels.

## Author contributions

**Conceptualization:** Huali Jin.

**Data curation:** Huali Jin, Chao Han.

**Formal analysis:** Huali Jin.

**Funding acquisition:** Huali Jin.

**Investigation:** Huali Jin.

**Methodology:** Huali Jin, Chao Han.

**Writing – original draft:** Huali Jin.

**Writing – review & editing:** Huali Jin.

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
