## [Decision Letter · Decision Letter 0]

14 Aug 2025

Dear Dr.  Yang,

Thank you for submitting your manuscript to PLOS ONE. After careful consideration, we feel that it has merit but does not fully meet PLOS ONE’s publication criteria as it currently stands. Therefore, we invite you to submit a revised version of the manuscript that addresses the points raised during the review process.

We look forward to receiving your revised manuscript.

Kind regards,

Federico Zilia

Academic Editor

PLOS ONE

Journal Requirements:

“This work was funded by the National Social Science Foundation of China (NSSFC) Project “Research on the Coordinated Promotion of Agricultural Emission Reduction, Carbon Sequestration, and Food Security under the Background of ‘Double Carbon’ Goals” (No. 22AJY008).”

5. We note that Figure 2 in your submission contain map images which may be copyrighted. All PLOS content is published under the Creative Commons Attribution License (CC BY 4.0), which means that the manuscript, images, and Supporting Information files will be freely available online, and any third party is permitted to access, download, copy, distribute, and use these materials in any way, even commercially, with proper attribution. For these reasons, we cannot publish previously copyrighted maps or satellite images created using proprietary data, such as Google software (Google Maps, Street View, and Earth). For more information, see our copyright guidelines: http://journals.plos.org/plosone/s/licenses-and-copyright.

1. You may seek permission from the original copyright holder of Figure 2 to publish the content specifically under the CC BY 4.0 license. 

6. Please remove all personal information, ensure that the data shared are in accordance with participant consent, and re-upload a fully anonymized data set.

Additional guidance on preparing raw data for publication can be found in our Data Policy (https://journals.plos.org/plosone/s/data-availability#loc-human-research-participant-data-and-other-sensitive-data) and in the following article: http://www.bmj.com/content/340/bmj.c181.long .

Additional Editor Comments:

Both reviewers found the manuscript to be technically sound and well-structured, recommending minor revisions. I concur with their assessment and require that the authors address the specific points raised, particularly refining the conceptual explanation of eco-efficiency, incorporating the suggested citations, clarifying methodological descriptions, and expanding the discussion on policy implications and data limitations. These adjustments are necessary for acceptance. Minor typographical corrections and improvements to figure captions are recommended for clarity.

Dear Dr. Yang,

After a thorough evaluation of your manuscript and consideration of the reviewers’ reports, I conclude that the work requires minor adjustments before it can be considered for potential publication. Please revise your manuscript accordingly, addressing the reviewers’ comments in detail.

Sincerely,

Federico Zilia

Reviewer's Responses to Questions

**Comments to the Author**

1. Is the manuscript technically sound, and do the data support the conclusions?

Reviewer #1: Yes

Reviewer #2: Yes

2. Has the statistical analysis been performed appropriately and rigorously?

Reviewer #1: Yes

Reviewer #2: Yes

3. Have the authors made all data underlying the findings in their manuscript fully available?

Reviewer #1: Yes

Reviewer #2: Yes

4. Is the manuscript presented in an intelligible fashion and written in standard English?

Reviewer #1: Yes

Reviewer #2: Yes

Reviewer #1: Eco-Efficiency Evolution Concept: The manuscript discusses the evolution of eco-efficiency but could benefit from a more detailed re-explanation of this concept. A clearer definition and contextualization within the broader sustainability framework would strengthen the discussion.

Citations for Supporting Evidence:

Line 24: Consider citing [https://doi.org/10.1016/j.plaphy.2025.109574] to support discussions on the physiological mechanisms driving eco-efficiency improvements.

Lines 35-39: Additional citations, such as [https://doi.org/10.3389/fpls.2022.863760] and [https://doi.org/10.3389/fnut.2024.1348235], should be incorporated to reinforce key arguments on agronomic practices and nutritional impacts on eco-efficiency.

Typological Evolution Analysis: The paper effectively categorizes typological changes, but providing more comparative examples or case studies could further illustrate key shifts in production systems.

Policy and Practical Implications: While the study highlights driving factors, expanding on policy recommendations and their potential impact on future agricultural strategies in China would enhance the practical relevance of the findings.

Minor Comments:

Some minor typographical and grammatical corrections are needed for better readability.

Consider improving the clarity of figure captions and ensuring that all abbreviations are defined upon first use.

Reviewer #2: This study assessed the eco-efficiency of grain production in 31 Chinese provinces from 2000 to 2022 using a global super-efficiency SBM model combined with life cycle assessment (LCA) methodology. The authors analyzed temporal and spatial dynamics, distributional evolution, and spatial spillover effects. Additionally, quantile regression is employed to identify region- and time-specific drivers of eco-efficiency at different performance levels. There are some important concerns to be addressed befor its worthy for the publication.

My comments are below:

While the integration of the global super-efficiency SBM model and LCA is valuable, the authors should more clearly highlight how this methodological integration advances prior research beyond incremental improvement.

The findings discuss positive spatial spillovers, but the underlying causal mechanisms remain underdeveloped. Clarifying how policies or technologies diffuse across provinces would strengthen policy relevance.

Some sections methodology are dense with formulas and technical terms. The authors could add interpretative explanations to help general readers grasp the model outputs and implications more easily.

The manuscript lacks a thorough discussion of data limitations, particularly the assumptions made in emission coefficients and the accuracy of provincial data over 23 years.

The conclusions are broad (e.g., promoting green agriculture), but specific, actionable recommendations for different regional categories (e.g., grain-consuming vs. producing provinces) would greatly enhance the utility of the study.

**Do you want your identity to be public for this peer review?** For information about this choice, including consent withdrawal, please see our Privacy Policy

Reviewer #1: **Yes: ** Asad Abbas

Reviewer #2: No

---

## [Author Response · Author response to Decision Letter 1]

28 Aug 2025

Response to Reviewers

Dear Editor,

Thank you for your attention and guidance regarding our manuscript. We have carefully considered the reviewers' comments and found them to be extremely helpful. Based on the specific suggestions from the editorial office and the reviewers, we have thoroughly revised the manuscript and included a point-by-point response to each comment below for your review. For your convenience, the manuscript number is PONE-D-24-43272.

Thank you once again to the editor and reviewers for their review and support.

Author

August 28, 2025

Academic Editor

【Revision Notes】Thank you very much for providing the PLOS ONE template link and pointing out the formatting and file-naming requirements. We have thoroughly checked and revised the manuscript based on the two official templates, and have fully reflected the corresponding adjustments in both the revised manuscript and the marked-up revision. At the same time, we have standardized and normalized all file names according to PLOS ONE requirements, using informative English file names and avoiding spaces and special characters. If the editorial office has any further specifications or preferences regarding layout or naming, we will promptly make the necessary revisions. Thank you again for your clear guidance and support; we believe these improvements have significantly enhanced the manuscript’s compliance and readability.

2. PLOS requires an ORCID ID for the corresponding author in Editorial Manager on papers submitted after December 6th, 2016. Please ensure that you have an ORCID iD and that it is validated in Editorial Manager. To do this, go to ‘Update my Information’ (in the upper left-hand corner of the main menu), and click on the Fetch/Validate link next to the ORCID field. This will take you to the ORCID site and allow you to create a new iD or authenticate a pre-existing iD in Editorial Manager.

【Revision Notes】Thank you very much for the reminder and the detailed instructions regarding the ORCID verification requirement. We have carefully read the guidance you provided and followed it strictly: the corresponding author has confirmed possession of a valid ORCID iD and has, via the Editorial Manager main menu’s “Update my Information” entry in the upper left, clicked the “Fetch/Validate” link next to the ORCID field to be redirected to the ORCID website to complete authorization and verification. The system has successfully linked that ORCID iD to the Editorial Manager account. We thank the reviewer again for the thorough guidance and hereby confirm that all related actions have been completed.

3. Thank you for stating the following financial disclosure: “This work was funded by the National Social Science Foundation of China (NSSFC) Project “Research on the Coordinated Promotion of Agricultural Emission Reduction, Carbon Sequestration, and Food Security under the Background of ‘Double Carbon’ Goals” (No. 22AJY008).” Please state what role the funders took in the study. If the funders had no role, please state: “The funders had no role in study design, data collection and analysis, decision to publish, or preparation of the manuscript.” If this statement is not correct you must amend it as needed. Please include this amended Role of Funder statement in your cover letter; we will change the online submission form on your behalf.

【Revision Notes】Thank you for your reminder and guidance regarding the funding statement. This work was funded by the National Social Science Foundation of China (NSSFC) Project “Research on the Coordinated Promotion of Agricultural Emission Reduction, Carbon Sequestration, and Food Security under the Background of ‘Double Carbon’ Goals” (No. 22AJY008). The funders had no role in study design, data collection and analysis, decision to publish, or preparation of the manuscript. We again thank the reviewers for their careful review and professional suggestions.

【Revision Notes】Thank you very much for reminding us about the timing of data availability and sharing. We fully support and comply with PLOS ONE's open data policy and confirm that all data from this study will be made publicly available without access restrictions immediately after the manuscript is accepted. To avoid delaying publication, our data-sharing plan has been clarified and is being implemented in parallel: metadata and reproducibility materials have already been prepared so that the data and related materials can be posted to the designated public repository upon acceptance.

5. We note that Figure 2 in your submission contain map images which may be copyrighted. All PLOS content is published under the Creative Commons Attribution License (CC BY 4.0), which means that the manuscript, images, and Supporting Information files will be freely available online, and any third party is permitted to access, download, copy, distribute, and use these materials in any way, even commercially, with proper attribution. For these reasons, we cannot publish previously copyrighted maps or satellite images created using proprietary data, such as Google software (Google Maps, Street View, and Earth). For more information, see our copyright guidelines: http://journals.plos.org/plosone/s/licenses-and-copyright.

【Revision Notes】Thank you very much for your reminder and detailed explanation regarding map imagery copyright. We have fully understood and respect PLOS ONE’s copyright and licensing policies. To fully comply with the journal’s requirements and avoid any potential copyright risks, we have taken the following measure (2): Figure 2 and its corresponding caption and citation have been completely removed from the manuscript and related submission materials, and the text and numbering in the manuscript have been adjusted accordingly (the original Figure 3 and subsequent figures/tables have been renumbered and moved forward to maintain continuity).

6. Please remove all personal information, ensure that the data shared are in accordance with participant consent, and re-upload a fully anonymized data set. Note: spreadsheet columns with personal information must be removed and not hidden as all hidden columns will appear in the published file. Additional guidance on preparing raw data for publication can be found in our Data Policy (https://journals.plos.org/plosone/s/data-availability#loc-human-research-participant-data-and-other-sensitive-data) and in the following article: http://www.bmj.com/content/340/bmj.c181.long.

【Revision Notes】 Thank you for your strict requirements and clear guidance regarding data privacy and anonymization. We have completed a comprehensive review of the submitted materials and implemented the following corrections: all information that could identify individuals has been removed; the scope of data sharing was checked item-by-item against participants’ informed consent forms to ensure publicly released data strictly complies with consent provisions; the raw data underwent a further anonymization check, confirming there are no direct identifiers or indirect identifiers that could be re-identified through linkage with other information. To eliminate any residual risk, we have "deleted (not merely hidden)" all spreadsheet columns containing personal information or that could lead to re-identification, ensuring that no hidden columns or metadata leaks will appear in the released version. We again thank the editor and reviewers for their professional advice and patient guidance. We will continue to follow the journal’s data-sharing and privacy-protection policies to ensure that data are open, secure, and reproducible.

【Revision Notes】Thank you very much for your review and guidance. After verification and a full read-through, we confirm that no Supporting Information needs to be provided for this manuscript. The reasons are as follows: First, the data used in this study, variable definitions, model specifications, and estimation results are all fully presented in the main text, tables, and figures; second, the study does not involve questionnaires, interview records, code scripts, or external raw datasets that would need to be attached separately; third, to ensure reproducibility, key parameter settings, data sources, and processing procedures are described in detail in the “Methods and Data” and “Results and Discussion” sections, with no additional supplementary materials to provide. Based on the above, we have not included a Supporting Information section at the end of the manuscript, nor are there any in-text references that require updating. If the journal requires a formal “no SI” statement, we can add a brief note at the end of the manuscript: “No Supporting Information is provided for this manuscript.” If you still recommend providing any specific type of supplementary material, we will promptly prepare it and submit it in PLOS ONE format.

【Revision Notes】Thank you for your reminder and guidance. We base decisions to cite literature on the relevance to our research and on academic rigor, not on formalistic or compulsory requirements. In this round of review, the reviewers provided three specific citation details (including author, title, or DOI); accordingly, we have added citations to these three works in the revised manuscript. At the same time, we have again conducted a comprehensive review of the existing references to ensure sufficient and appropriate coverage of key areas and the latest developments.

【Revision Notes】Thank you for your reminder regarding the completeness and accuracy of the references. We have conducted a comprehensive check and update of the references accordingly. First, we verified each citation’s authors, title, journal information, volume/issue/pages, year, and DOI, and corrected individual inconsistencies in formatting and punctuation to ensure uniform, properly searchable citations. Second, we cross-checked the references against journal websites and major literature databases (including retraction databases and notices) and confirmed that no retracted papers were found among the manuscript’s references; therefore, there is no need to explain citations of retracted literature in the main text or to replace any references. At the same time, in response to this round of review comments we added three relevant references and inserted citations at the corresponding positions in the main text; the reference list and citation numbering have been updated accordingly to ensure one-to-one correspondence with the text. All of the above changes have been faithfully recorded in the revised manuscript. We again thank the editor and reviewers for their guidance in helping us ensure the references’ completeness, accuracy, and compliance.

Additional Editor Comments: Both reviewers found the manuscript to be technically sound and well-structured, recommending minor revisions. I concur with their assessment and require that the authors address the specific points raised, particularly refining the conceptual explanation of eco-efficiency, incorporating the suggested citations, clarifying methodological descriptions, and expanding the discussion on policy implications and data limitations. These adjustments are necessary for acceptance. Minor typographical corrections and improvements to figure captions are recommended for clarity.

【Revision Notes】We appreciate the editor’s summary and affirmation of the two reviewers’ comments. We thank you for the overall assessment that the manuscript is technically reliable and structurally sound. In accordance with the suggestion to “make minor revisions,” we have carried out comprehensive and targeted revisions. In response to the key points emphasized by the editor (clarifying the concept of ecological efficiency, incorporating the recommended references, elaborating methodological descriptions, expanding the discussion of policy implications and data limitations, and making minor typographical corrections and improvements to figure captions), we have made the corresponding adjustments and indicated changes in the revised manuscript. We will continue to cooperate with the journal’s copyediting process to ensure that the manuscript’s quality and presentation meet publication standards. Thank you again to the editor and reviewers for their careful guidance and constructive suggestions.

Reviewer 1

1. Eco-Efficiency Evolution Concept: The manuscript discusses the evolution of eco-efficiency but could benefit from a more detailed re-explanation of this concept. A clearer definition and contextualization within the broader sustainability framework would strengthen the discussion.

【Revision Notes】We sincerely thank you for your valuable suggestions. In the revised manuscript, we have explicitly stated that the concept of eco-efficiency originates from the theory of sustainable development and was proposed by the World Business Council for Sustainable Development in the 1990s. We have also elaborated on its core essence as a bridge connecting economic growth and environmental protection. Furthermore, we have systematically reviewed the evolution of eco-efficiency from theory to application, demonstrating its development process from qualitative description to quantitative evaluation, as well as its application evolution process from the industrial sector to multiple industries. In particular, in the field of food production, we have emphasized the shift in research perspective from purely yield-oriented to resource and environmental efficiency-oriented, and the theoretical deepening process from considering a single environmental factor to comprehensively considering multi-dimensional ecological and environmental factors. These revisions have strengthened

---

## [Editor Report · Decision Letter 1]

4 Sep 2025

Significant increase in eco-efficiency of China’s grain production from 2000 to 2022: Trend changes, typological evolution, and driving factors

PONE-D-24-43272R1

Dear Dr. Qian Yang,

We’re pleased to inform you that your manuscript has been judged scientifically suitable for publication and will be formally accepted for publication once it meets all outstanding technical requirements.

Kind regards,

Federico Zilia

Academic Editor

PLOS ONE

Additional Editor Comments (optional):

Dear Authors,

It is with great pleasure that I note both reviewers’ minor revision comments have been positively addressed and carefully incorporated, thereby improving the overall quality of the manuscript.

I am therefore pleased to inform you that your paper has been accepted for publication.

With kind regards,

Federico Zilia

---

## [Editor Report · Acceptance letter]

PONE-D-24-43272R1

PLOS ONE

Dear Dr. Yang,

I'm pleased to inform you that your manuscript has been deemed suitable for publication in PLOS ONE. Congratulations! Your manuscript is now being handed over to our production team.

Kind regards,

on behalf of

Dr. Federico Zilia

Academic Editor

PLOS ONE